# Core genes can have higher recombination rates than accessory genes within global microbial populations

**Asher Preska Steinberg[1], Mingzhi Lin[1], Edo Kussell[1,2]\***

[1]Department of Biology, New York University, New York, United States; [2]Department of Physics, New York University, New York, United States

**Abstract** Recombination is essential to microbial evolution, and is involved in the spread of antibiotic resistance, antigenic variation, and adaptation to the host niche. However, assessing the impact of homologous recombination on accessory genes which are only present in a subset of strains of a given species remains challenging due to their complex phylogenetic relationships. Quantifying homologous recombination for accessory genes (which are important for niche-specific adaptations) in comparison to core genes (which are present in all strains and have essential functions) is critical to understanding how selection acts on variation to shape species diversity and genome structures of bacteria. Here, we apply a computationally efficient, non-phylogenetic approach to measure homologous recombination rates in the core and accessory genome using >100,000 whole genome sequences from *Streptococcus pneumoniae* and several additional species. By analyzing diverse sets of sequence clusters, we show that core genes often have higher recombination rates than accessory genes, and for some bacterial species the associated effect sizes for these differences are pronounced. In a subset of species, we find that gene frequency and homologous recombination rate are positively correlated. For *S. pneumoniae* and several additional species, we find that while the recombination rate is higher for the core genome, the mutational divergence is lower, indicating that divergence-based homologous recombination barriers could contribute to differences in recombination rates between the core and accessory genome. Homologous recombination may therefore play a key role in increasing the efficiency of selection in the most conserved parts of the genome.

**\*For correspondence:**
edo.kussell@nyu.edu

**Competing interest:** The authors declare that no competing interests exist.

## Editor's evaluation

Homologous recombination is an important process driving the evolution of bacterial genomes. This paper addresses how the rate of homologous recombination varies between core and accessory genes. The authors quantify recombination using an approach based on the decline of linkage between polymorphic sites over distance. The analysis indicates that core genes can be under higher rates of homologous recombination than accessory genes, which is an interesting observation that contrasts with the prevailing wisdom. Together, it suggests that homologous recombination may play a key role in increasing the efficiency of selection in conserved components of the genome.

## Introduction

Intrinsic to bacterial genome evolution is the process of recombination (*Fraser et al., 2007*; *Hanage, 2016*; *Smith, 1991*; *Thomas and Nielsen, 2005*), which can enhance the ability of microbes to adapt to their environment (*Arnold et al., 2022*; *Maiden, 1998*; *van der Woude and Bäumler, 2004*). Recombination in bacteria occurs in two forms. One is 'homologous recombination', in which fragments of

DNA taken up from the environment are incorporated into homologous sites in the genome resulting in allele replacement and no gain or loss of chromosomal DNA (recently this has also been referred to as 'allele transfer' [*Arnold et al., 2022*]). The second is 'nonhomologous recombination' or 'horizontal gene transfer' (HGT), in which DNA fragments are inserted or removed from the genome, changing the amount of chromosomal DNA and potentially leading to gene gain or loss events (*Arnold et al., 2022*; *Hanage, 2016*). Both of these processes obfuscate clonal signals and confound phylogenetic analyses (*Feil et al., 2001*; *Guttman and Dykhuizen, 1994*; *Spratt et al., 2001*), and recombination is so ubiquitous in certain bacteria that whether or not bacterial genome evolution can truly be characterized as tree-like is the subject of much debate (*Creevey et al., 2004*; *Doolittle, 1999*; *Sakoparnig et al., 2021*; *Spratt et al., 2001*). Recombination rates vary greatly both between and within different bacterial species (*Castillo-Ramírez et al., 2012*; *Chaguza et al., 2016*; *Chewapreecha et al., 2014*; *Croucher et al., 2013*; *Hanage, 2016*; *Hanage et al., 2009*), and this variance is attributed to a complex interplay of selective pressure, molecular mechanisms, and ecological factors which have yet to be disentangled. Along the genome, recombination rates vary among specific genes (i.e., recombination 'hotspots') (*Didelot et al., 2012*; *Everitt et al., 2014*) and between gene classes such as 'informational genes' (those involved with transcription/translation) and 'operational genes' (those involved with biosynthesis, metabolism, and regulatory functions) (*Jain et al., 1999*; *Novick and Doolittle, 2019*).

This inter- and intragenomic variation in recombination rates has presented difficulties with using individual genes (e.g., 16S rRNA) to define genotypic clusters and species, and approaches such as multilocus sequence typing (MLST) have been developed to address this (*Hanage et al., 2005*; *Hanage et al., 2006a*; *Maiden et al., 1998*). Multilocus approaches attempt to resolve this issue by concatenating sequences from several housekeeping genes, thereby minimizing the chance that local inter- and intraspecific recombination at a single locus distorts our interpretation of clonal relationships. These housekeeping genes are part of the 'core' genome, which is found in nearly all strains of a given species and is thought to be less prone to recombination events (at least in the form of HGT) compared to the collections of genes found in only subsets of strains of a given species known as the 'accessory' genome (*Daubin et al., 2002*; *Lan and Reeves, 2001*; *McInerney et al., 2017*).

Quantifying the variation of homologous recombination rates across core and accessory genes is important for understanding of how selection and recombination interact to shape sequence diversity, as core genes are thought to be under strong, purifying selection (*Bohlin et al., 2014*; *Bohlin et al., 2017*; *den Bakker et al., 2013*; *McInerney et al., 2017*; *Moulana et al., 2020*) whereas many accessory genes are thought to be important for niche specialization and under positive or diversifying selection (*Azarian et al., 2020*; *McInerney et al., 2017*; *Vernikos et al., 2015*). Moreover, characterizing homologous recombination rates in niche-adaptive accessory genes will have far-reaching implications for understanding how bacteria adapt to stresses such as antibiotics, host immune responses, and nutrient conditions (*Didelot et al., 2016*; *Povolo and Ackermann, 2019*; *Seifert, 1996*; *van der Woude and Bäumler, 2004*). However, measuring homologous recombination rates across accessory genes has been stymied by the frequent gene loss and gain events that they experience, which cause difficulties in applying phylogeny-based approaches to infer homologous recombination rates (*Arnold et al., 2018*; *Ansari and Didelot, 2014*; *Croucher et al., 2015*; *Didelot et al., 2010*; *Didelot et al., 2012*; *Didelot and Falush, 2007*; *Didelot and Wilson, 2015*; *Iranzo et al., 2019*; *Lefébure and Stanhope, 2007*; *Marttinen et al., 2012*; *Mostowy et al., 2017*). For this reason, variation in recombination rates has primarily been studied across core genes (*Didelot et al., 2012*; *Everitt et al., 2014*; *Jain et al., 1999*). Whereas previous studies indicate that the core genome experiences fewer HGT events (*Daubin et al., 2002*; *Haegeman and Weitz, 2012*; *Lan and Reeves, 2001*; *Lobkovsky et al., 2013*; *Wolf et al., 2016*), direct comparisons of homologous recombination rates between the core and accessory genomes are currently lacking.

Here, we determine how homologous recombination rates differ between core and accessory genes. We focus on *Streptococcus pneumoniae*, which frequently engages in homologous recombination, has been extensively sequenced (*Chaguza et al., 2016*; *Chewapreecha et al., 2014*; *Croucher et al., 2013*; *Hanage, 2016*; *Hanage et al., 2009*), and is clinically relevant (*O'Brien et al., 2009*). For these reasons, *S. pneumoniae* provides a key case study for quantifying variation in homologous recombination rates across the core and accessory genome and, by analyzing additional species, we determine the generality of these trends across several bacterial species.

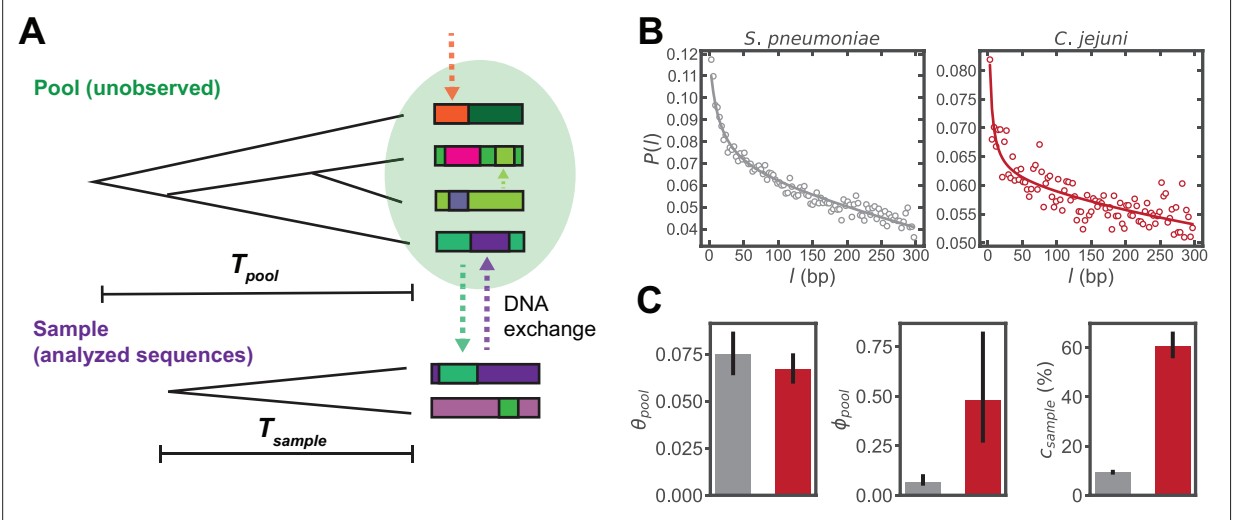

**Figure 1.** Inferring parameters of homologous recombination from whole genome sequences (WGS). (**A**) Schematic depicting exchange of homologous DNA fragments between the analyzed sequences ('Sample') and a larger, unobserved reservoir of bacterial genomes ('Pool'). The coalescence times of the pool and sample are denoted as $T_{pool}$ and $T_{sample}$, respectively. (**B**) Example correlation profiles of synonymous substitutions for the core genes in samples consisting of 568 *S. pneumoniae* and 2215 *C. jejuni* WGS (further description of these datasets are given in the following sections). (**C**) Recombination parameters inferred from fitting the profiles shown in panel C to a population genetics model (see Materials and methods). Left to right: the pool's mutational divergence ($\theta_{pool}$), the pool's recombinational divergence ($\phi_{pool}$), and the sample's recombination coverage ($c_{sample}$). Error bars are 95% bootstrap confidence intervals (see Materials and methods). Colors correspond to panel B. $\theta_{pool}$ and $\phi_{pool}$ have units of $\mathrm{bp}^{-1}$ and $c_{sample}$ is given as the percentage of genomic sites that have recombined.

## Results

### Inferring recombination parameters for analyzed samples and unobserved gene pools using correlated substitutions

To infer the parameters of homologous recombination, we apply and extend the *mcorr* method (*Lin and Kussell, 2017*), an approach that avoids phylogenetic reconstruction by using a coalescent-based population genetics model with recombination to capture the statistics of large-scale sequencing data (see *Lin and Kussell, 2019* for details). We define a 'sample' to be a set of lineages with mean coalescence time $T_{sample}$, which mutate and exchange homologous DNA fragments with a much larger, unobserved 'pool' of sequences having mean coalescence time $T_{pool}$ (*Figure 1A*). By fitting the model to the data, we infer recombination parameters of both the sample (i.e., the specific set of sequences used for analysis) and its pool; sample construction and collection are discussed in the next section. We obtain the pool's mutational divergence, $\theta_{pool} \equiv 2\mu T_{pool}$, the pool's recombinational divergence, $\phi_{pool} \equiv 2\gamma T_{pool}$, the sample's recombination coverage, $c_{sample}$, and the mean recombined fragment size, $\bar{f}$ (where $\mu$ and $\gamma$ are the synonymous substitution rate and recombination rate, respectively). Together, $\theta_{pool}$ and $\phi_{pool}$ estimate the number of synonymous substitutions and recombination events that have occurred per site since coalescence of the pool, and $c_{sample}$ estimates the fraction of the genome that has been replaced by homologous recombination with the pool since coalescence of the given sample. Unlike the synonymous substitutions, which we assume to be largely neutral, recombination events may be due to selective pressure or neutral drift.

The model predicts the conditional probability of a difference at genome site $i + l$ given a difference at site $i$, which we refer to as the 'correlation profile', $P(l)$, where $l$ is the distance between sites in basepairs (bp). We take a set of whole genome sequences (WGS) to constitute a sample, and use alignments of protein coding (CDS) regions to measure profiles of synonymous substitutions for all possible sequence pairs, yielding an average profile for the sample. For each pair of sequences in the sample at genomic position $i$, a binary variable $\sigma_i$ is assigned 1 for a difference or 0 for identity. The correlation profile is given by $P(l) \equiv P(\sigma_{i+l} = 1 | \sigma_i = 1)$, where $i$ is restricted to third position sites of codons. While the profiles are computed using gene alignments, recombined fragments may span multiple genes, and the mean fragment size in the coalescent model can take values larger than

the length of single genes. Fitting the model to these correlation profiles yields the parameters of homologous recombination described above (see *Lin and Kussell, 2019* and Materials and methods for details of model and fitting); flat profiles indicate a lack of recombination, while in the presence of recombination $P(l)$ has a monotonic decay with $l$, and declines more rapidly with increasing recombination (*Lin and Kussell, 2017*). Example profiles and inferred parameters are shown in *Figure 1B, C* using sets of *S. pneumoniae* and *Campylobacter jejuni* WGS; these sequences are part of the larger datasets used in the analysis that follows. Thus, the method infers parameters of both the sample and the pool of sequences with which it has recombined without using or inferring any phylogenetic relationships, offering a key advantage in determining homologous recombination rates across accessory genes.

## Homologous recombination rates are higher in the core versus accessory genes for *S. pneumoniae*

To study recombination across the core and accessory genome of a given bacterial species, we analyze a wide range of samples from the global microbial population. In our model, each 'sample' is composed of collections of genomes from nature; they are statistical samples from a larger, unobserved bacterial population or gene pool. By examining a diverse set of samples from a given species, we are able to infer the distribution of recombination parameters for their unobserved gene pools. However, as many genomes are collected from specific geographic regions or individual hospitals (*Chaguza et al., 2016*; *Chewapreecha et al., 2014*; *Pelton et al., 2007*), a sample can be artificially overweighted by a single collection site accounting for many genomes. Therefore, to obtain a more meaningful set of samples for a given species, and ameliorate some of the compositional biases inherent in nonrandom sampling, we use sequence clusters obtained by clustering WGS solely based on sequence similarity using the average linkage algorithm (*Wheeler and Kececioglu, 2007*) with the pairwise synonymous diversity ($d_s$) as the distance metric. Such approaches are widely used to define well-resolved genotypic clusters within bacterial populations without the inference of phylogenetic relationships (*Hanage et al., 2006a*). We then analyze single clusters or pairs of clusters to infer the distribution of pool parameters. For each cluster or cluster pair, we infer both the divergences of the unobserved gene pool with which they recombine (i.e., $\theta_{pool}$ and $\phi_{pool}$), as well as the amount of recombination that has taken place within the set of analyzed sequences (i.e., the cluster or cluster pair) since coalescence ($c_{sample}$). For single clusters, we fit correlation profiles measured by averaging over all of a given cluster's sequence pairs, which yields the parameters of the unobserved pool that the cluster interacts with. For cluster pairs, we fit correlation profiles measured by averaging exclusively over sequence pairs consisting of one sequence from each of the two clusters. By analyzing all possible pairs of clusters, we use diverse sets of sequences as our samples, yielding a parameter distribution that accounts for the full range of potential interactions between different sequences, regardless of their distance in sequence space and agnostic to population structure or sampling biases.

To sample the global population of *S. pneumoniae*, we used the *PubMLST* database (see *Supplementary file 5*) which includes WGS of strains from across the world. For each WGS, we aligned all sequencing reads to a reference genome to create an alignment of all CDS regions (see Materials and methods for details). We then measured $d_s$ across all genes (i.e., core and accessory) for each strain pair, clustered based on these distances to make a dendrogram, and split the sequences into flat clusters, where no two sequences within a cluster were more distant than the 10th percentile of pairwise distances, which corresponded to $d_s = 0.015$ (*Figure 2A*, Materials and methods for details; mean $d_s$ values within and between clusters for core and accessory genes are given in *Supplementary file 3*). This resulted in 44 major clusters (where a major cluster has >100 strains) which we used as our samples (946 total cluster pairs; *Figure 2B* shows an example correlation profile from a cluster), and we inferred recombination parameters for the core (defined here as present in >95% of strains) and accessory genes of each of the samples and the unobserved pools with which they recombine (*Figure 2C–E*).

We found that the core and accessory genes of *S. pneumoniae* have distinct distributions of $\theta_{pool}$ and $\phi_{pool}$ (*Figure 2C, D*), and the median of $\theta_{pool}$ is lower in core versus accessory genes, indicating that core genes are less mutationally diverged than accessory genes. Moreover, we observed that $\phi_{pool}$ is higher for the core genes, implying that core gene pools have experienced a greater number of recombination events per site compared to accessory genes (*Figure 2D*). We examined this further

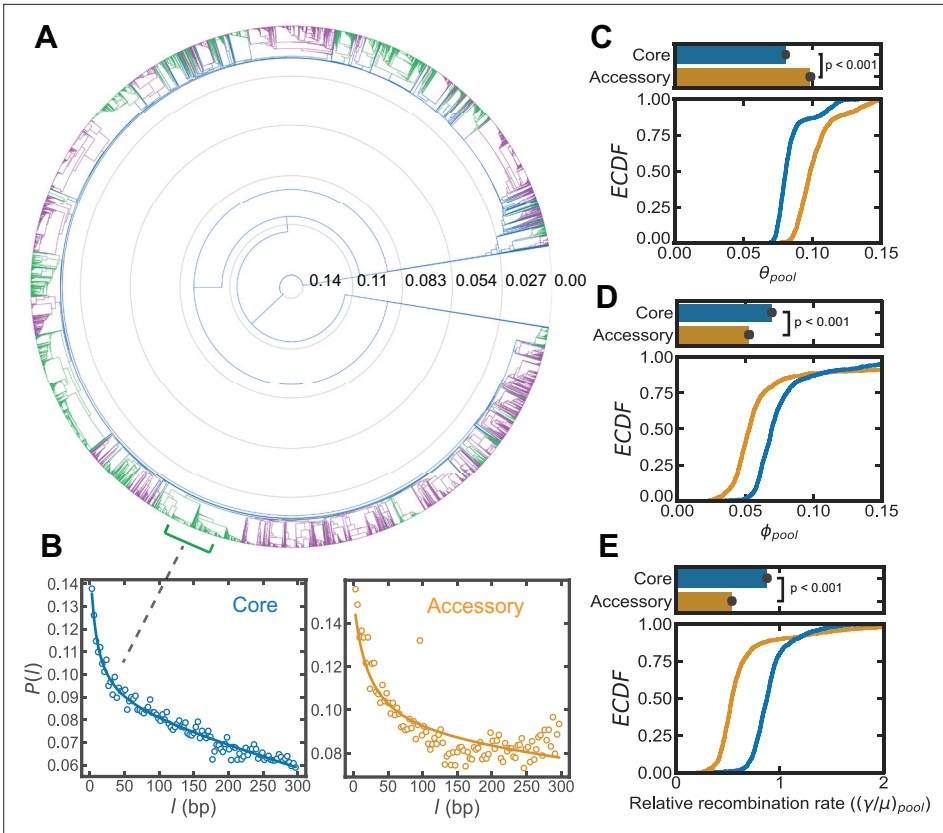

**Figure 2.** Inference of recombination parameters for the core and accessory genome of *S. pneumoniae*. (**A**) Dendrogram resulting from hierarchical clustering of 26,599 whole genome sequences (WGS) from the *PubMLST* genome collection for *S. pneumoniae* using the average linkage algorithm. The dendrogram was cut at the 10th percentile of all measured pairwise distances ($d_s \sim 0.015$) yielding a discrete set of flat clusters. Here, we analyzed the 44 major clusters (those with >100 sequences) which resulted from this cut, encompassing 24,097 strains. In the dendrogram, alternating colors delineate adjacent clusters. (**B**) Correlation profiles measured across the core and accessory genes of a *S. pneumoniae* sequence cluster. Open-faced circle is the profile measured from sequencing data and solid line is the fit to the population genetics model. (**C–E**) Distributions of the pool's mutational and recombinational divergence ($\theta_{pool}$ and $\phi_{pool}$, shown in panels C and D, respectively) and the relative recombination rate of the pool (($\gamma/\mu$)$_{pool}$, shown in panel E). For C–E, the bottom plot depicts empirical cumulative distribution functions (*ECDF*s) for each parameter and the top plots depict the medians of the bottom plot, where error bars are 95% bootstrap CIs created by sampling the distributions with replacement ($n = 1000$). Each step in the ECDF corresponds to a pool recombination parameter inferred from a correlation profile measured over a sequence cluster or pair of clusters like that shown in panel B. Core genes are defined as genes found in >95% of strains. We used model selection with the Akaike information criterion to ensure that each profile was well fit (see Materials and methods for details). $\theta_{pool}$ and $\phi_{pool}$ have units of $\mathrm{bp}^{-1}$ and ($\gamma/\mu$)$_{pool}$ is unitless. Two-sided p-values were calculated using the Wilcoxon signed-rank test and were $p = 1.9\mathrm{e}{-}159$, $3.9\mathrm{e}{-}53$, and $3.5\mathrm{e}{-}99$ for panels C–E, respectively.

The online version of this article includes the following source data and figure supplement(s) for figure 2:

**Source data 1.** List of SRA accession numbers corresponding to the raw reads of genomes used from NCBI.

**Figure supplement 1.** Recombination coverage and average size of a recombined fragment for *S. pneumoniae*.

**Figure supplement 2.** Recombination rate distributions for *S. pneumoniae* using different core gene cutoff thresholds.

**Figure supplement 3.** Comparison of recombination rates for the actual core and accessory genome versus an artificially 'thinned' core genome for *S. pneumoniae*.

**Figure supplement 4.** Comparison of distributions of recombination parameters for the core and accessory genome of *S. pneumoniae* with different alignment methods.

using the relation $\phi_{pool}/\theta_{pool} = \gamma/\mu$, which removes the dependence on coalescence time to yield the relative recombination rates of the unobserved gene pools (from hereon we will append the subscript *pool* to indicate this, i.e., $(\gamma/\mu)_{pool}$). The inferred distribution of $(\gamma/\mu)_{pool}$ shows that for *S. pneumoniae*, the core genome has a higher relative recombination rate than the accessory genome (*Figure 2E*), again indicating that core genes recombine more frequently. While the inferred distribution of recombination coverage for the samples initially suggests that the accessory genome may recombine more (*Figure 2—figure supplement 1A*), this can be reconciled by considering the mean size of the recombined fragments ($\bar{f}$), which is higher for the accessory genes (*Figure 2—figure supplement 1B*). Taken together, we find that while accessory genes incorporate larger fragments, the core genes of the sampled genomes experience more recombination events on a per site basis (*Figure 2—figure supplement 1C*).

We tested the robustness of these results in several ways. As there are various thresholds used to define 'core' genes (*Livingstone et al., 2018*; *McInerney et al., 2017*; *Page et al., 2015*; *Vernikos et al., 2015*), we tested how the parameter distributions shifted with different thresholds and found similar trends (*Figure 2—figure supplement 2*). To determine whether the difference in the abundance of the core and accessory genes (i.e., how many strains a gene appears in) influences our inference of recombination parameters, for each cluster we artificially 'thinned' the core genome by replacing each accessory gene sequence with a randomly chosen core gene sequence. This yielded a thinned core genome where each core gene had the same number of sequence pairs as the accessory genome. We found that this thinned core genome showed similar recombination trends as the actual core genome (*Figure 2—figure supplement 3*).

Aligning raw reads to a reference genome to build consensus genomes allows for access to a much larger set of WGS, as there are comparably fewer whole genome assemblies for *S. pneumoniae*, which are typically a requirement for reference-free alignment (*Ding et al., 2018*; *Lin and Kussell, 2019*; *Page et al., 2015*) (as of the time of this analysis NCBI GenBank had 81 complete genome assemblies for *S. pneumoniae* versus the ~26,000 WGS accessed through PubMLST). However, a potential drawback of aligning reads to a single reference genome is that a subset of accessory genes will be missed. New approaches for aligning to reference graphs appear promising in this regard, yet remain computationally expensive for large numbers of reads (*Colquhoun et al., 2021*). We therefore sought to address this issue by testing how the parameters changed when more accessory genes were included by building a reference pangenome from all the genome assemblies in NCBI GenBank using *Roary* (*Page et al., 2015*) and aligned all the raw reads from PubMLST to this (see Materials and methods). Pangenome analysis suggests that this number of genomes should encompass the majority of genes in the *S. pneumoniae* pangenome (*Donati et al., 2010*). While the pangenome alignment included more genes (6200 versus 2018 genes), we found very similar trends for the inferred recombination parameters to those inferred using a single reference alignment (*Figure 2—figure supplement 4*).

To assess the generality of the trends in recombination parameters between the core and accessory genome, we analyzed 11 additional microbial species (encompassing >100,000 genomes including those already analyzed for *S. pneumoniae*). We used the same procedure described above to cluster sequences and measure correlation profiles for clusters and cluster pairs, and inferred distributions of recombination rates for the unobserved pools for these species. Pairing the inferred recombination rates of core and accessory genomes for each pool, we examined the effect sizes for each species using Cohen's *d* statistic (*Cohen, 1988*; *Nakagawa and Cuthill, 2007*) to determine the magnitude and direction of the effect (*Figure 3A*; *Supplementary file 1*). We found that, in line with our observations in *S. pneumoniae*, certain species (e.g., *Escherichia coli*, *Shigella flexneri*, *Neisseria meningitidis*, and *C. jejuni*) showed markedly higher recombination rates for core genes, whereas others showed slightly higher recombination rates in the core genome. *Helicobacter pylori* had higher recombination rates in the accessory genome which were statistically significant, yet the magnitude of the effect was small (Cohen's *d* = 0.18). Among the nine species for which Cohen's *d* was nonzero (within 95% confidence intervals) all but *H. pylori* showed higher recombination rates (*d* < 0) in the core versus accessory genome. Similarly, of the 10 species with significantly different median recombination rates in core and accessory genomes (within 95% confidence intervals) (*Supplementary file 1*), all but *H. pylori* and *N. gonorrhoeae* showed higher median recombination rates in the core versus accessory genome. Different metrics thus yield largely consistent results regarding differences in recombination rates between core and accessory genomes across species. Lastly, to understand if core gene pools

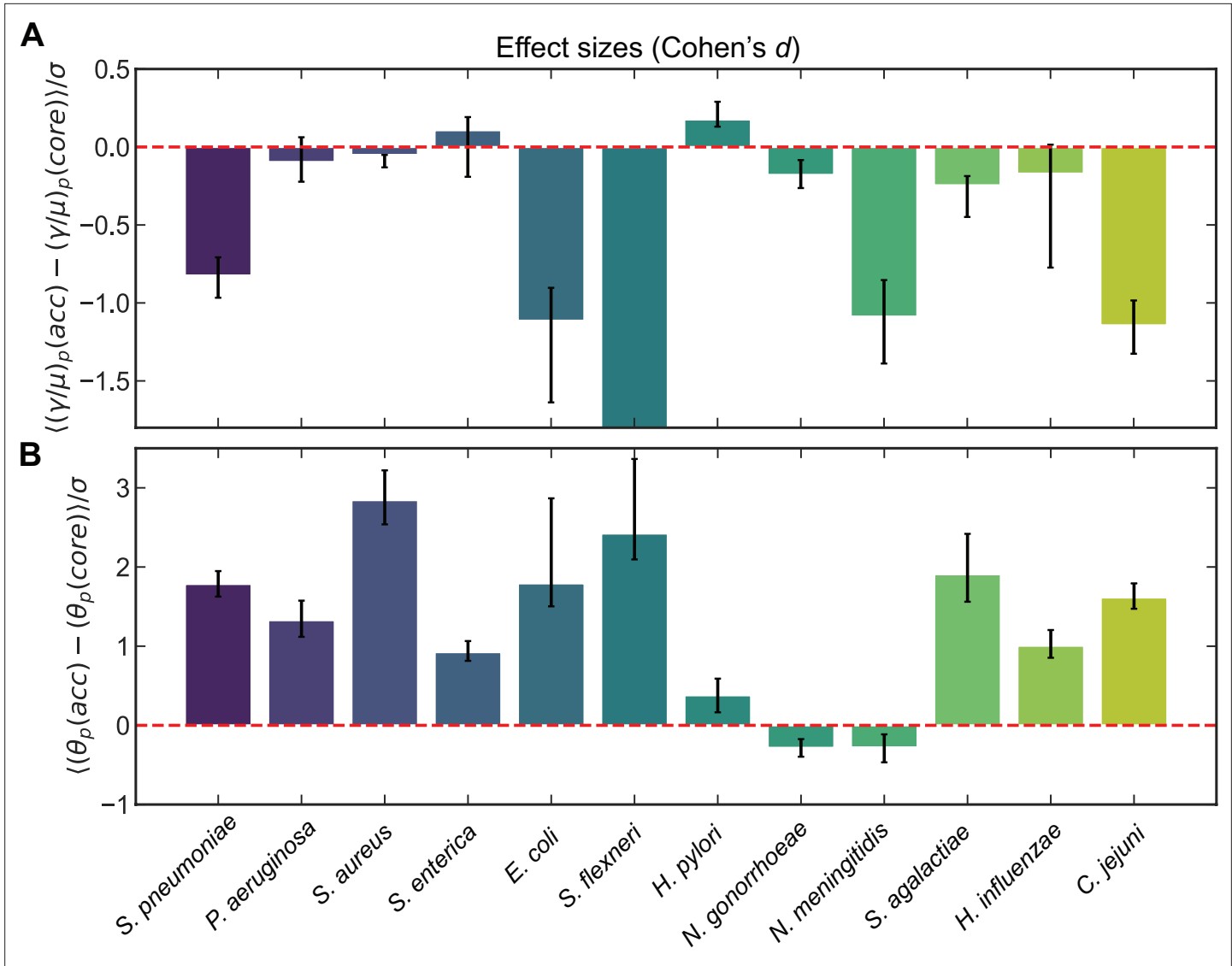

**Figure 3.** Effect sizes (Cohen's *d*) for recombination rates and mutational divergence for *S. pneumoniae* and 11 additional microbial species. Effect sizes (Cohen's *d*) for (**A**) the relative recombination rate of the pool $((\gamma/\mu)_p)$ and (**B**) the mutational divergence of the pool $(\theta_p)$ for core/accessory genome pairs for individual clusters and pairs of clusters for *S. pneumoniae* and 11 additional microbial species. Cohen's *d* for paired samples was calculated as the mean paired difference $(\langle X_p(acc) - X_p(core)\rangle$, where $X = (\gamma/\mu)_p$ or $\theta_p)$ divided by the standard deviation of paired difference $(\sigma)$. Error bars are 95% bootstrap CIs, calculated by sampling the distributions with replacement 10,000 times. All effect sizes are listed in *Supplementary files 1 and 2*, as well as the medians of the distributions and the results of the Wilcoxon signed-rank test for each species. For A, the effect size and 95% bootstrap CI for *S. flexneri* are −25.5 and [−34.4,−21.1], respectively. When model selection was performed with Akaike information criterion (AIC) (described in Materials and methods) if part of a core/accessory pair was poorly fit, the paired sample was excluded. Full species names, number of strains, number of clusters, and mean synonymous diversity within and between clusters are reported for each species in *Supplementary file 3*. Legends of supplementary material.

The online version of this article includes the following source data and figure supplement(s) for figure 3:

**Source data 1.** Lists of SRA accession numbers corresponding to the raw reads of genomes used from NCBI.

**Figure supplement 1.** Dependence of recombination rates on gene frequency.

are less diverged than accessory pools as we had found for *S. pneumoniae*, we also assessed differences in $\theta_{pool}$ between the core and accessory genome for these same 12 species (*Figure 3B*; *Supplementary file 2*). We found that a majority of species had accessory gene pools which were markedly more diverged than core genes, and the few exceptions (e.g., *N. gonorrhoeae* and *N. meningitidis*) had small effect sizes.

## Gene frequency as a recombination barrier in the core and accessory genome

One testable hypothesis consistent with our observation that core genes generally undergo more homologous recombination when compared with accessory genes is that the least frequent genes experience less recombination because they have fewer recombination partners. In other words, gene frequency could act as a recombination barrier. We tested this by binning the genes for the microbial species in *Figure 3* into four gene frequency classes consistent with prior delineations of the pangenome as follows ($q$ is frequency across strains) (*Koonin and Wolf, 2012*; *Page et al., 2015*): 'cloud' genes ($q \leq 15\%$), 'shell' genes (divided into two segments, $15\% < q \leq 55\%$, $55\% < q \leq 95\%$), and core genes ($q > 95\%$). We then inferred the distributions of recombination parameters within each gene class for each of the 12 microbial species (*Figure 3—figure supplement 1A, B*). We found that certain species displayed trends consistent with a gene frequency barrier, particularly for the recombination coverage (*Figure 3—figure supplement 1A*). When considering the recombination rates (displayed as empirical cumulative distribution functions here because of disparate rates across and within species), *C. jejuni*, *S. agalactiae*, and *P. aeruginosa* showed trends most consistent with this hypothesis. For some of the exceptions in which cloud genes showed high recombination rates (e.g., *N. gonorrhoeae*, *N. meningitidis*, *E. coli*, and *S. flexneri*), this could be related to the high gene replacement rate experienced by rare genes such as 'ORFans' (*Wolf et al., 2016*), or diversifying selection experienced by these genes. Overall, we see a modest positive correlation with both recombination coverage and rates when considering the 50th and 25th percentiles of the matched parameter distributions for all species (*Figure 3—figure supplement 1C, D*).

## Discussion

There is appreciable interest in understanding why microbes have pangenomes and how different parts of the genome have evolved (*Lobkovsky et al., 2013*; *McInerney et al., 2017*; *Vernikos et al., 2015*; *Wolf et al., 2016*). While it is known that homologous recombination plays a key role in shaping the genome (*Fraser et al., 2007*; *González-Torres et al., 2019*; *Hanage, 2016*; *Iranzo et al., 2019*; *Thomas and Nielsen, 2005*), large-scale analysis of this aspect of genome evolution has been limited. This was due to both computational bottlenecks and a reliance on phylogenetic methods (*Arnold et al., 2018*; *Ansari and Didelot, 2014*; *Croucher et al., 2015*; *Didelot et al., 2010*; *Didelot and Falush, 2007*; *Didelot and Wilson, 2015*; *Marttinen et al., 2012*; *Mostowy et al., 2017*), the latter of which hindered determination of recombination parameters for accessory genes, whose phylogenies are challenging to ascertain. Here, we expanded our non-phylogenetic, computationally efficient *mcorr* framework to overcome these obstacles, and present a detailed analysis of the variation in recombination rates between core and accessory genes for *S. pneumoniae*, along with several other microbial species.

We found that despite the high gene replacement rates and sequence variability experienced by some accessory genes (*Iranzo et al., 2019*; *Kuenne et al., 2013*; *Kuhn et al., 2006*; *Wolf et al., 2016*), core genes experience markedly higher recombination rates than the accessory genes in *S. pneumoniae*. In 8 out of 12 microbial species, we observe higher recombination rates in core genes relative to accessory genes. In a subset of microbial species, the magnitude of the effect is large (e.g., *S. pneumoniae*, *E. coli*, *S. flexneri*, *N. meningitidis*, and *C. jejuni*). While some work has previously shown housekeeping (*Vos and Didelot, 2009*) and core genes (*Lin and Kussell, 2017*; *Park and Andam, 2020*) can experience extensive recombination consistent with our results, our work offers a direct, quantitative comparison between the core and accessory genomes spanning multiple species and >100,000 sampled genomes. In certain species, mechanisms that give rise to higher recombination rates in the core genome are known; for example, some members of the *Neisseriaceae* (e.g., *N. meningitidis* and *N. gonorrhoeae*) and *Pasteurellaceae* families (e.g., *H. influenzae*) have DNA uptake sequences and uptake signal sequences accumulated in their core genomes which promote homologous recombination (*Frye et al., 2013*; *Treangen et al., 2008*).

We found that in *S. pneumoniae*, core gene pools had lower mutational divergence relative to accessory gene pools. Furthermore, we found that for the majority of the species we analyzed this was the case, and 9 out of 12 species analyzed had large effect sizes indicating core gene pools have lower mutational divergence (Cohen's $d$ = 0.92–2.8). Previous studies indicate that the core genome

is under strong, purifying selection (**Bohlin et al., 2014**; **Bohlin et al., 2017**; **den Bakker et al., 2013**; **McInerney et al., 2017**; **Moulana et al., 2020**), and it is well known that purifying selection reduces mean coalescence times (**Charlesworth et al., 1993**; **Nicolaisen and Desai, 2012**). An additional consideration is that some accessory genes are used for niche specialization (**Azarian et al., 2020**; **McInerney et al., 2017**; **Vernikos et al., 2015**); if different alleles of the same accessory gene are adaptive to different niches, this could result in allelic diversity due to balancing selection, which would increase coalescence times of accessory genes. The lower value of $\theta_{pool}$ in core genes is therefore consistent with shorter coalescence times of core relative to accessory genes. While another possible explanation could involve codon usage biases (**Plotkin and Kudla, 2011**) causing differences in synonymous substitution rates between core and accessory genes, the selective coefficients for this process are tiny ($s \sim 1/N_e$ **Hershberg and Petrov, 2008**); $N_e = 3 \times 10^8$ for *S. pneumoniae* (**Bobay and Ochman, 2018**) and thus effectively neutral over the timescales that we can observe in these data (i.e., coalescence times of *S. pneumoniae* samples and pools). Accordingly, evidence from long-term microbial evolution experiments has shown that on average the synonymous substitution rate is fairly homogeneous across the core and accessory genomes (**Maddamsetti et al., 2015**; **Maddamsetti et al., 2017**). Therefore, we attribute the difference in $\theta_{pool}$ between core and accessory genes to the difference in their coalescence times, which is consistent with core genes being under stronger purifying selection, and additionally with the possibility that a subset of accessory genes are under diversifying selection. We note that because higher levels of recombination in core genes are expected to reduce the effects of background selection (**Charlesworth et al., 1993**), the relative reduction in $\theta_{pool}$ observed in core versus accessory genes of *S. pneumoniae* and other species likely underestimates the difference in strength of purifying selection acting on these gene classes.

Experiments in numerous species show that homologous recombination rates decay with increasing sequence divergence (**Fraser et al., 2007**; **Majewski, 2001**; **Majewski et al., 2000**; **Vulić et al., 1997**; **Zawadzki et al., 1995**), and the role of this recombination barrier in bacterial speciation has been analyzed (**Falush et al., 2006**; **Hanage et al., 2006b**). This effect is primarily ascribed to the ubiquity of RecA and mismatch repair systems, both of which are essential for recombination and whose molecular mechanisms depend on sequence similarity. Consistent with these experimental and analytical results, the higher recombination rates and lower pool divergence we observe here for the core genome of *S. pneumoniae* and several other microbial species may suggest that the increased homology of the core genes reduces the barrier for homologous recombination in this part of the genome. This is also in line with modeling, which has suggested that homologous recombination slows sequence divergence in the core genome by homogenizing this part of the genome, whereas accessory genes do not have the opportunity to be homogenized as they are subject to continual gene gain and loss (**Iranzo et al., 2019**). Thus, we propose that an evolutionary feedback loop may operate in certain species of bacteria: purifying selection acting on core genes causes higher levels of homology at these loci, this increases their homologous recombination rates, which in turn enables more efficient purifying selection by breaking linkages between genes, effectively minimizing Hill–Robertson interference and hitchhiking effects (**Barton, 1995**; **Barton, 2010**; **Felsenstein and Yokoyama, 1976**) across the core genome. Further investigations are needed to examine whether such an evolutionary mechanism can act to fine-tune recombination rate variation across microbial genomes. Moreover, our work suggests that another recombination barrier for accessory genes may be related to their lower abundance in the population, as we found that the levels of homologous recombination are correlated with gene frequency in a subset of species.

In summary, we have presented a quantitative analysis of the variation in recombination rates between core and accessory genomes. As these two parts of the genome are under different forms of selective pressures, this work contributes broadly to our understanding of how selection and recombination act to shape diversity. The expansion of this approach to more specific gene classes could lead to a better understanding of how selective pressure and homologous recombination interplay to shape niche-adaptive genes such as those involved with drug resistance and antigen variation, for which allele shuffling due to homologous recombination can play a central role in their evolution (**Bowler et al., 1994**; **Maiden, 1998**; **Seifert, 1996**).

## Materials and methods
### Data and code availability

- Lists of SRA accession numbers corresponding to the raw reads used to build the multi-sequence alignments analyzed here are included as source data. All SRA files, reference genomes, and complete genome assemblies are available through NCBI. All sequence collections used are listed in *Supplementary file 5*. For the PubMLST sequence collections, PubMLST was used to identify WGS (by filtering for strains in the 'Genome Collection' of each species where the sequence length is at least that of the reference genome), then the raw reads were downloaded from NCBI using their SRA numbers. Accession numbers for reference genomes used for each microbial species are also listed in *Supplementary file 5*.
- All original code has been deposited at GitHub and is publicly available (links provided throughout Methods).

### Description of coalescent-based population genetics model with recombination

The details of the model derivation and parameters are given in the 'Supplementary Notes' of *Lin and Kussell, 2019*. Here, we provide the model equations necessary for the fitting procedure used in this work.

The measured synonymous diversity of the sample, $d_s$, is a linear combination of the pool diversity, $d_p \equiv d(\theta_p)$ (due to recombination), and the clonal diversity of the sample, $d(\theta_s)$, as follows:

$$d_s = c_s\, d(\theta_p) + \left(1 - c_s\right) d(\theta_s),\tag{1}$$

where $c_s$ is the recombination coverage in the sample, and $d(\theta)$ is given by the classic population genetics expression for heterozygosity (*Wakeley, 2009*) (see table below). The predicted form of the correlation profile is

$$P(l) = c_{s,0}(l)\, d(2\theta_s) + c_{s,1}(l)\, d_p + c_{s,2}(l)\, Q_p(l)\,/\,d_s\,,\tag{2}$$

where the functional forms of $c_{s,i}(l)$ ($i = 0, 1, 2$) and $Q_p(l)$ are given in the table below. As described in *Lin and Kussell, 2019*, we can reexpress *Equation 2* in terms of the three parameters $\theta_s$, $\phi_s$, and $\bar{f}$, determined by fitting the profiles. From these, the pool parameters are obtained using the following relations: $\theta_p = \dfrac{d_s(1+ \phi_s w\bar{f}+\theta_s\tilde{a})-\theta_s}{(1-d_s\tilde{a})(\phi_s w\bar{f}+\theta_s\tilde{a})-d_s\tilde{a}}$ and $\phi_p = \theta_p\phi_s/\theta_s$ ; see table below for values of constants $\tilde{a}$ and $w$.

| Term | Expression | Description |
|---|---|---|
| $\theta_s$ or $\theta_p$ | | Mutational divergence of the sample or pool |
| $\phi_s$ or $\phi_p$ | | Recombinational divergence of the sample or pool |
| $\bar{f}$ | | Mean fragment size of homologous recombination |
| $l$ | | Distance (bp) between two loci |
| The following constants are used below: $\tilde{a} = 4/3$ and $w = 2/3$. | | |
| $d(\theta)$ | $\dfrac{\theta}{1+\theta\tilde{a}}$ | Synonymous diversity (heterozygosity) for divergence $\theta$ |
| $c_s$ | $\dfrac{\phi_s w\bar{f}}{1+\theta_s\tilde{a}+\phi_s w\bar{f}}$ | Recombination coverage of the sample |
| $c_{s,1}(l)$ | $\dfrac{2\phi_s wl}{1+2\theta_s\tilde{a}+\phi_s w(\bar{f}+l)}$ | Probability that the most recent event was a recombination affecting only one locus |
| $c_{s,2}(l)$ | $\dfrac{\phi_s w(\bar{f}-l)}{1+2\theta_s\tilde{a}+\phi_s w(\bar{f}+l)}$ | Probability that the most recent event was a recombination affecting both loci |
| $c_{s,0}(l)$ | $1 - c_{s,1} - c_{s,2}$ | Probability that the most recent event at either locus was not a recombination |
| $Q_p(l)$ | $2\left(\dfrac{\theta_p}{1+\theta_p\tilde{a}}\right)^2\left(\dfrac{1+\theta_p\tilde{a}+\phi_p wl}{1+2\theta_p\tilde{a}+2\phi_p wl}\right)$ | Probability of a difference at both loci in the pool |

## Calculation of correlation profiles

For a given sample of aligned sequences, we measured the substitution profile, $\sigma_i(k, g)$, for each pair of sequences $k$, at each position along each gene $g$, where $\sigma_i(k, g) = 1$ for a difference and $\sigma_i(k, g) = 0$ for identity. In all calculations below, we only consider positions that are third position sites of codons yielding synonymous substitutions. We computed the pairwise synonymous diversity of each gene $g$ as:

$$d_s(g) = \left\langle \overline{\sigma_i(k, g)} \right\rangle \tag{3}$$

where the bar denotes averaging over all sequence pairs $k$ and the bracket denotes averaging over all positions $i$. The joint probability of synonymous substitutions for two sites separated by a distance $l$ was calculated for each gene as:

$$Q_s(l, g) = \left\langle \overline{\sigma_i(k, g)\sigma_{i+l}(k, g)} \right\rangle \tag{4}$$

We averaged these over all genes to obtain the *sample diversity*, $d_s = (1/n_g) \sum_g d_s(g)$, and the joint probability of substitutions at two sites in the sample, $Q_s(l) = (1/n_g) \sum_g Q_s(l, g)$, where $n_g$ is the total number of genes. The correlation profile is then given by $P(l) = Q_s(l) / d_s$.

For calculations of *within-pool* parameters, all possible sequence pairs within a cluster were considered. This was done with the original command-line (CLI) program *mcorr-xmfa* program in the *mcorr* package which takes as input a single extended multi-fasta (XMFA) file. For calculations of *between-pool* parameters using pairs of clusters, only sequence pairs where each sequence was from a different cluster were considered. This was done with the CLI program created for this paper called *mcorr-xmfa-2clades*, which uses two XMFA files (one from each sequence cluster) as inputs. These can be found in the *mcorr* GitHub repository: https://github.com/kussell-lab/mcorr.

## Fitting of correlation profiles and model selection using Akaike information criterion

The basic fitting procedure used was previously described in **Lin and Kussell, 2019**. We used the python package LMFIT (**Newville et al., 2014**) version 0.9.7 (https://lmfit.github.io/lmfit-py/) to fit $P(l)$ to the analytical form given in **Equation 2**; in this work, we used the least-squares minimization with Trust Region Reflective method instead of the default Levenberg–Marquardt algorithm and increased the maximum number of function evaluations from $10^4$ to $10^6$. As described in the text, as an additional test of goodness of fit, we compared the fit of the data with **Equation 2** where all parameters vary freely (which we refer to as the 'full-recombination model'), to the 'null-recombination model' in which we set $c_{s,1} = c_{s,2} = 0$, which yields $P(l) = d(2\theta_s) = 2\theta_s/(1 + 2\theta_s \tilde{a})$, that is, a constant correlation profile which is fit by taking the average over $l$ of the measured values, yielding a single parameter, $\theta_s$. We perform model selection between the null- and full-recombination models by evaluating the Akaike information criterion (AIC) for each model, then computing the Akaike weight for each model, which can be roughly interpreted as the probability that a given model yields the best prediction of the data (**Bois, 2020**; **Wagenmakers and Farrell, 2004**). The AIC was computed using LMFIT using $AIC = n \ln(\chi^2/n) + 2N_v$, where $n$ is the number of data points, $N_v$ is the number of parameters being varied in the fitting, and $\chi^2 = \sum_{i=1}^{n} \left[ Resid_i \right]^2$ is the sum of the squared residuals for each data point . The Akaike weight for model $j \in \{\varnothing, r\}$ can then be calculated as:

$$w_j = \frac{e^{-0.5\left(AIC_j - AIC_{min}\right)}}{e^{-0.5\left(AIC_\varnothing - AIC_{min}\right)} + e^{-0.5\left(AIC_r - AIC_{min}\right)}}$$

where $AIC_\varnothing$ and $AIC_r$ denote the AIC values for the null- and full-recombination models, respectively, and $AIC_{min} = \min\{AIC_\varnothing, AIC_r\}$. The evidence ratio, which corresponds to the likelihood of one model being favored over the other in terms of minimizing Kullback–Leibler discrepancy (**Wagenmakers and Farrell, 2004**), can be computed as $w_r/w_\varnothing$, where $w_r$ and $w_\varnothing$ are the Akaike weights of the full- and null-recombination models, respectively. Fits to correlation profiles where $w_r / w_\varnothing < 100$ were not included in the analyses presented in the main text. We also did not include fits to correlation profiles which yielded unphysical parameter values ($\theta_{pool} < 0$, $\phi_{pool} < 0$), or extreme values of recombination coverage where parameter fitting becomes less reliable due to too few recombination events

($c_{sample} < 1\%$) or the over-fitting of a flat profile ($c_{sample} > 99\%$). The CLI program *mcorrFitCompare* was built for this study to fit correlation profiles to both models and can be found in the *mcorr* GitHub repository: https://github.com/kussell-lab/mcorr.

## Clustering using pairwise synonymous diversity

In this study, we extended the *mcorr* method to measure the synonymous diversity separately for each sequence pair, denoted by $d_{pair}$. This is the same as the calculation of $d_s$ described in 'Calculation of correlation profiles' without averaging over sequence pairs $k$. The CLI program *mcorr-pair* measures $d_{pair}$ for all possible sequence pairs from an XMFA file. We also built a CLI program called *mcorr-pair-sync*, which calculates a subset of all pairwise diversities from a given set of sequences. We primarily relied on the latter program, as it allows for parallelization of jobs on a high-performance computing cluster. The CLI programs *mcorr-dm* and *mcorr-dm-chunks* were built to collect outputs from *mcorr-pair* and *mcorr-pair-sync*, respectively, and write them to a square distance matrix for use with standard clustering algorithms. Lastly, we built a program (*clusterSequences*) relying on the python *scipy* package to cluster sequences using the unweighted pair group method with arithmetic mean or average linkage method with $d_{pair}$ as the distance metric. All necessary code to perform these calculations can be found in the github repository: https://github.com/kussell-lab/mcorr-clustering.

## Generation of multisequence alignments for core and accessory genes using WGS

For all collections of WGS (*Supplementary file 5*), with the exception for *H. pylori* (where the multisequence alignment was already assembled), we used reference-guided alignment to build consensus genomes from raw reads for each sequence, then extracted the CDS regions of each gene to make extended multi-fasta (XMFA) files. For the collections from the PubMLST database, we identified WGS by filtering for all sequences in the 'Genome Collection' for a given organism which had lengths greater than or equal to the length of the reference genome. We then exported tables which included the corresponding SRA numbers for each strain, and downloaded the reads from NCBI using the corresponding SRA. For the other sequence collections used, we used their associated NCBI Bioproject ID to obtain the SRA numbers for each strain, and downloaded reads in the same way. Reads were mapped against a reference genome (listed in *Supplementary file 5* for each microbial species) using SMALT (version 0.7.6; https://www.sanger.ac.uk/tool/smalt-0/), and consensus genomes were created using SAMtools mpileup (*Li et al., 2009*). We then extracted the alignments of CDS regions from these consensus genomes using genomic coordinates provided from the general feature format (GFF) file of the reference genome to create an XMFA file of CDS regions for each gene. We filtered out any gene alignments where the gene sequence was $\geq 2\%$ gaps. We then measured the percentage of the entire strain collection which had each gene (based on presence/absence of a gene sequence) to determine if a gene should be considered core or accessory. Code to perform this analysis is provided in the GitHub repository: https://github.com/kussell-lab/ReferenceAlignmentGenerator.

## Generation of multisequence alignment to pangenome reference genome for *S. pneumoniae*

We downloaded all complete genome assemblies from NCBI for *S. pneumoniae* as of December 18, 2020 (81 genomes), and used Prokka (*Seemann, 2014*) to reannotate the assemblies and generate general feature format version 3 (GFF3) files for use with Roary (*Page et al., 2015*). Roary was then used to generate a multi-FASTA for each gene CDS region in the pangenome (which we refer to as the 'pangenome reference'). We aligned reads to this pangenome reference in the same manner described in 'Generation of multisequence alignments for core and accessory genes using WGS', then collected each gene alignment to create an XMFA file for the pangenome. We split the XMFA into files for the core and accessory genome by measuring the percentage of strains which had each gene. In this case, because alignment quality was not as high as when we aligned reads to a single reference genome (as described in the main text), we counted a gene as present if there was only a partially aligned sequence for the gene, and absent if there was no aligned sequence. Code to perform this analysis can be found in the following GitHub repository: https://github.com/kussell-lab/PangenomeAlignmentGenerator.

## Generation of artificially 'thinned' core genome alignment for *S. pneumoniae*

To create the artificially 'thinned' core genome alignment described in the main text, we took the core and accessory genome alignments for *S. pneumoniae* generated as described in 'Generation of multisequence alignments for core and accessory genes using WGS', and we replaced each of the accessory gene alignments for each sequence cluster with randomly selected core gene alignments from the same cluster until all accessory gene alignments were replaced. A CLI program called *ReduceCoreGenome* is used to generate these thinned core genome alignments (https://github.com/kussell-lab/ReferenceAlignmentGenerator).

## Statistical analysis

For the 95% bootstrap CIs appearing in *Figure 1C*, the same procedure was used as in *Lin and Kussell, 2019*. In brief, bootstrap replicates of the set of genes were created by resampling the list of all genes in the genome with replacement, then recalculating $d_s$, $Q_s$, and $P(l)$ for each replicate. We then performed the same fitting procedure which was done on the actual dataset to infer recombination parameters on each of the bootstrap replicates to generate 95% confidence intervals. The resampling was done 1000 times.

For all other 95% bootstrap CIs, the procedure used for bootstrapping can be found in the figure legends of the corresponding figures. The process of model selection using AIC is described in 'Fitting of correlation profiles and model selection using Akaike information criterion'. The number of major clusters and sequences used for each microbial species is given in *Supplementary file 3*.

## Acknowledgements

This work was supported in part by NIH grant R01-GM-097356. Asher Preska Steinberg is a Simons Foundation Awardee of the Life Sciences Research Foundation. We gratefully acknowledge Nobuto Takeuchi for useful discussions and feedback on the manuscript; the New York University (NYU) high-performance computing cluster for resources, and its staff for technical support.

## Additional information

### Funding

| Funder | Grant reference number | Author |
| --- | --- | --- |
| National Institutes of Health | R01-GM097356 | Edo Kussell |
| Simons Foundation | Simons Foundation Awardee of the Life Sciences Research Foundation | Asher Preska Steinberg |

The funders had no role in study design, data collection, and interpretation, or the decision to submit the work for publication.

### Author contributions

Asher Preska Steinberg, Conceptualization, Resources, Data curation, Software, Formal analysis, Validation, Investigation, Visualization, Methodology, Writing - original draft, Writing - review and editing; Mingzhi Lin, Conceptualization, Software, Investigation; Edo Kussell, Conceptualization, Resources, Formal analysis, Supervision, Funding acquisition, Validation, Investigation, Methodology, Writing - original draft, Project administration, Writing - review and editing

### Author ORCIDs

Asher Preska Steinberg http://orcid.org/0000-0002-8694-7224
Edo Kussell http://orcid.org/0000-0003-0590-4036

**Decision letter and Author response**
Decision letter https://doi.org/10.7554/eLife.78533.sa1
Author response https://doi.org/10.7554/eLife.78533.sa2

## Additional files

### Supplementary files

• Supplementary file 1. Effect sizes, results of Wilcoxon signed-rank test, and medians of the recombination rate distributions for *S. pneumoniae* and 11 additional microbial species. Values listed in the table were computed for core/accessory genome pairs for individual clusters and pairs of clusters. Effect sizes for the relative recombination rate ($(\gamma/\mu)_{pool}$) were calculated using Cohen's $d$ for paired samples using the mean paired difference ($\langle (\gamma/\mu)_p (acc) - (\gamma/\mu)_p (core) \rangle$) divided by the standard deviation of paired differences ($\sigma$). Two-sided p values were calculated using the Wilcoxon signed-rank test to compare the paired distributions of relative recombination rates for the core and accessory genome within each microbial species. The null hypothesis is that the distribution of paired differences is symmetric about zero (where $X^i(acc) - X^i(core)$ is the paired difference for recombination parameter $X$ inferred for the core and accessory genome of cluster or cluster pair $i$). The columns $(\gamma/\mu)_{pool}(acc)$ and $(\gamma/\mu)_{pool}(core)$ correspond to the median relative recombination rate of the accessory and core genome over all clusters and pairs of clusters. Values displayed in brackets are 95% bootstrap CIs calculated by sampling the distributions with replacement 10,000 times.

• Supplementary file 2. Effect sizes, results of Wilcoxon signed-rank test, and medians of the mutational divergence distributions for *S. pneumoniae* and 11 additional microbial species. Values listed in the table were computed using the pool's mutational divergence ($\theta_{pool}$) for core/accessory genome pairs for individual clusters and pairs of clusters. This table is identical to *Supplementary file 1* except the statistics displayed here were calculated using $\theta_{pool}$.

• Supplementary file 3. Description of the microbial species analyzed in *Figures 2 and 3* and *Supplementary files 1 and 2*. Column descriptions are as follows: 'species' and 'full name' give the abbreviated species name used in the main text and full species name, respectively, 'major clusters' gives the number of sequence clusters analyzed, 'total strains aligned' gives the number of consensus genomes (or whole genome sequences) which were made by aligning raw reads to the reference genome, 'total strains in major clusters' gives the number of strains included in the major clusters (and therefore included in the analysis of recombination parameters), and 'min strains per cluster' gives the minimum number of strains a cluster had to have to be designated as a major cluster. The minimum cluster size was lowered for smaller strain collections as follows (where $s$ is the number of strains in the collection): for $s > 10{,}000$ in the collection the minimum cluster size was 100, for $10{,}000 > s > 5000$ the minimum was 50 (*S. enterica* was on the border, so we took the minimum to be 50), for $5000 > s > 1000$ the minimum was 25, and for $s < 1000$ the minimum was 10. The last four columns give the average synonymous diversity ($d_s$) within clusters and between clusters (shown as mean ± st. dev.). 'Acc' stands for accessory.

• Supplementary file 4. Results of Friedman test comparing matched distributions for each of the nine microbial species shown in *Figure 3—figure supplement 1*. Two-sided p values were calculated using the Friedman test to compare the matched distributions of the normalized recombination rate ($\gamma/\mu_{pool}$) and recombinational coverage ($c_{sample}$) for all four gene frequency bins within each microbial species shown in *Figure 3—figure supplement 1*. Null hypothesis in this case is that the measurements have been drawn from the same distribution.

• Supplementary file 5. Identifying information for sequencing data used in study. Source of data and corresponding identifiers for all sequencing data used in this study.

• MDAR checklist

### Data availability

Lists of SRA accession numbers corresponding to the raw reads used to build the multisequence alignments analyzed in this manuscript are included as Figure 2—source data 1 and Figure 3—source data 1. All SRA files, reference genomes, and complete genome assemblies are available through NCBI. All sequence collections used are listed in Supplementary file 5. For the PubMLST sequence collections, PubMLST was used to identify whole genome sequences (by filtering for strains in the 'Genome Collection' of each species where the sequence length is at least that of the reference genome),

then the raw reads were downloaded from NCBI using their SRA numbers. Accession numbers for reference genomes used for each microbial species are also listed in Supplementary file 5. All original code has been deposited at GitHub and is publicly available at https://github.com/kussell-lab (copies archived at swh:1:rev:4adea90557f1e592123e073b46a859d879143a2a, swh:1:rev:1d34860183a5d-d55332e08edf9503c4ce6daebdf, swh:1:rev:49bf399a9d1ceae722c0c8c4aeb8376f2644d40f, and swh:1:rev:2228e8ee2df7339d28ef8b9107381d2e4767ac11).

The following previously published dataset was used:

| Author(s) | Year | Dataset title | Dataset URL | Database and Identifier |
|---|---|---|---|---|
| Thorell K | 2018 | Data from: Rapid evolution of distinct Helicobacter pylori subpopulations in the Americas | https://doi.org/10.5061/dryad.8qp4n | Dryad Digital Repository, 10.5061/dryad.8qp4n |

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
