## [Editor Report]

Homologous recombination is an important process driving the evolution of bacterial genomes. This paper addresses how the rate of homologous recombination varies between core and accessory genes. The authors quantify recombination using an approach based on the decline of linkage between polymorphic sites over distance. The analysis indicates that core genes can be under higher rates of homologous recombination than accessory genes, which is an interesting observation that contrasts with the prevailing wisdom. Together, it suggests that homologous recombination may play a key role in increasing the efficiency of selection in conserved components of the genome.

---

## [Decision Letter]

**Decision letter after peer review:**

[Editors’ note: the authors submitted for reconsideration following the decision after peer review. What follows is the decision letter after the first round of review.]

Thank you for submitting the paper "Core genes can have higher recombination rates than accessory genes within global microbial populations" for consideration by *eLife*. Your article has been reviewed by 2 peer reviewers, and the evaluation has been overseen by a Reviewing Editor and a Senior Editor. The following individual involved in review of your submission has agreed to reveal their identity: Daniel Falush (Reviewer #3).

Comments to the Authors:

We are sorry to say that, after consultation with the reviewers, we have decided that this work will not be considered further, at least in this version, for publication by *eLife*. This said, as you will see from the reviewers' comments, all recognise the importance of the subject and the need for new approaches to understand evolutionary forces shaping pan and core genomes. Your work is very much a step in the right direction. If you find it possible to deal with the requests made by the referees, we would be willing to consider a fresh submission.

Specifically, on the problematic side, is seeming discord between the major claims of the paper and the data supporting these claims. This may in part stem from lack of clarity concerning methodological aspects, but as both referees point out, there are reasons for concern. And, importantly, for this work to appeal to a broad readership, it is necessary for the work to be generally accessible. Additionally, the work would benefit greatly from a more nuanced placement of the subject within context of an extensive and relevant literature. This would particularly aid motivation for the study.

*Reviewer #2:*

Homologous recombination is a key process driving the evolution of bacterial genomes, and the rate at which this occurs varies widely between different species. Whilst this variation is still not completely understood it is certainly due to a complex combination of mechanistic and ecological drivers. In this paper Steinberg and Kussel ask a slightly different question by the addressing the variation in intra-genomic rate of homologous recombination, and specifically between core and accessory genes. The use of core genes for phylogenetic and population-level inference (for example in multi-locus sequence typing) is predicated on the view that the genes encoding essential 'housekeeping' functions are under high levels of purifying selection, and are therefore assumed not to be impacted by high levels of recombination from diverged sources that might act to accelerate diversification. In contrast, accessory genes, which are more likely to encode niche-specific traits involving interactions with the external environment, may become more selectively favoured from homologous recombination leading to increased diversification. It is important to critically re-evaluate this model as evidence accumulates, both for understanding how selection shapes the genome and populations, but also for understanding the utility of different parts of the genome for phylogenetic and population genetics inference.

Even before the concept of the pan and core genome was established, varying rates of recombination between different classes of genes was a subject of active research. Jain et al. (Proc Natl Acad Sci U S A 1999 Mar 30;96(7):3801-6. doi: 10.1073/pnas.96.7.3801) speculated over 20 years ago, that 'operational' (metabolic) genes experience higher rates of transfer than 'informational' genes (those involved in DNA processing) because the latter were more likely to be involved in complex protein-protein interactions (the so-called 'complexity hypothesis'). Together with the underlying rational of MLST, and the prior use of 16s rRNA as a universal marker, there remains a common wisdom that the most conserved, ubiquitous and essential genes are also those least likely to undergo recombination. For a recent discussion of the implications of the complexity hypothesis see Novick, A., Doolittle, W.F.. Biol Philos 35, 2 (2020). https://doi.org/10.1007/s10539-019-9727-6

Whilst it is unfortunate that none of this broad perspective is elaborated in the paper, Steinberg and Kussel address this question on a large-scale (10 species, >100,000 genomes), and note that in many (but not all) species accessory genes tend to undergo lower rates of homologous recombination than core genes, although this trend is far from universal. To measure recombination they use an approach called mcorr which gauges recombination on the basis of the steepness of the decline in linkage between polymorphic sites as a function of genome distance. Their model allows them to distinguish between mutational and recombinational divergence, as well as estimating the percentage of each gene sequence that has been impacted by recombination. Their results are a mixed bag, with variation between species and between the different parameters within single species. However, overall the suggestion is that accessory genes may experience less homologous recombination than core genes. This trend is consistent with arguments (some of which are expressed by the authors) relating to the homogenising effect of recombination between close parental sequences, and how, in this case, recombination can reinforce purifying rather than directional selection.

While the question is an important one, and the use of large genome sequences to address it is laudable, I sense that there are two many factors at play over this broad scale to draw any firm conclusions about general trends. My main concerns with the analyses are two-fold – first it is carried out on single gene alignments which severely limits the range over which recombination can be detected and quantified (this would not be so critical if all homologous recombination was by localised transformation, but transduction and conjugation also play major roles). My second concern lies in the use of a single reference genome for each species. For species with very large pan-genomes, such as *E. coli*, this effectively means that the majority of accessory genes (those that happen to be absent in the reference) will not be included. The authors do acknowledge this, and present an analysis based on a composite reference, but they reject this approach due to the poor alignments that result. This is the perennial problem with working with accessory genes – but a reference free method for detecting SNPs in accessory genes has recently been (eg Colquhoun, R.M., Hall, M.B., Lima, L. et al. Pandora: nucleotide-resolution bacterial pan-genomics with reference graphs. Genome Biol 22, 267 (2021)). There are additional potential complications relating to how the different datasets have been sampled that have not been acknowledged. In sum, the analysis should be of interest to the field as it does demonstrate the potential for addressing this fundamental question using large WGS datasets, but my sense is that the authors would have been better served to analyse one or two species in detail, taking into account all possible confounders, rather than spread themselves too thinly across such diverse datasets.

General: the paper would be vastly improved by a broader review of the literature and a more clear motivation for the study. The introduction is frankly a bit waffly.

Abstract:

The observation that some species consist of overlapping gene pools (or even one single gene pool) to a degree simply reflects inconsistencies in our demarcation of species (for example S. flexneri is really just a clone of *E. coli*, and N. gonorrhoeae is really just a clone of N. meningitidis).

Line 25 the role of population structure in genome evolution – I'm not sure about the causal direction here – it seems to make more sense the other way around to me.

Line 29 – it has been established for a long time that core genes are under strong purifying selection. Claiming this as news somehow detracts from the aspects of the paper that are really novel.

Line 43 – the ability to quantify rates in different parts of the genome hasn't been limited, but has been going on for a long time. For an example, see Nat Commun 2014 May 23;5:3956. doi: 10.1038/ncomms4956.

As I understand the model, it is based on the decay of linkage over distance (Figure 1C,D) but I have struggled a bit with how you move from this to the data you present in the other figures – it would be reassuring to us non-modellers if you could show some more examples of this decay to understand more clearly what the results you present are really based on.

Are recombination events, like synonymous SNPs, also assumed to be neutral?

Line 120 – explain w/n.

Line 136 – meaning 'no two sequences within a cluster'?

Figure 2B – why not separate points for within and between clusters?

Figure S2 – why exclude the marginal plots?

Line 194 – There is certainly an effect of codon bias on substitution rate – the fact the experimental evolution data hasn't picked this up is more a comment on these experiments.

Line 264 – the P values presented in S table 1 are two-signed – which ones indicate higher rates of recombination in the core genes?

Figure 4 – 60% of the core gene sequences in *S. enterica* have experienced recombination? This seems a lot -how does this figure (and other species specific data) compare with what is in the literature? Could the authors comment on the fact that in *E. coli* (and S. flexneri) the strict core genes show less recombination than the cloud core.

Line 322 'complicates' not 'complexifies'.

Line 332 – 340 this model isn't really novel.

*Reviewer #3:*

This paper presents interesting results comparing core and accessory genome evolution.

The headline result, if correct, is both important and reasonably easy to understand; namely that in Pneumococci and other bacterial species, on average homologous recombination rates (or more precisely the effect of recombination in reassorting sequence diversity) are higher in core genome elements, while diversity is higher in accessary genome elements.

This is an interesting result and is buttressed by showing some dependence on gene frequencies. It is not entirely surprising, since core genes have more opportunities for recombination due to being present in every strain, while accessory genome elements have more opportunity for being imported from other species. Nevertheless, a quantitative result is important enough to be newsworthy, even though it does not do an enormous amount to elucidate the mechanisms responsible for the difference. This would require more explicitly phylogenetic methods or a more in-depth analysis of the difference between species, which is currently superficial.

However, I am not sure I am persuaded by the second part of the result, since looking at supplementary figure 4, it is not obvious to me that there is a clear trend for recombination parameters, core and accessory genomes seem to give very inconsistent results in different species, for which no real explanations are given.

There is one important technical check that should be carried out, which is that core genes should be artificially thinned to match the frequency of accessory gene elements and then the analysis repeated, to see whether this has any effect on in the inference. Indeed, because the alignment method is shown to have a considerable effect on the inference, especially of recombination parameters (supplementary figure 2) this thinning would be ideally done before alignment, although I recognize that this would be time-consuming to implement.

Figure 3 is presented as the results from 12 species but it’s really hard to judge what this figure actually tells us. Agglomorating different species into single graphs seems meaningless and makes it impossible to see what is actually going on with individual species.

Aside from questions about to what extent the data really supports the headline result, my main problem with the paper is that the analysis underlying the result and in particular the meaning of the figures is difficult to understand. As it stands, I do not think it is suitable for a broad readership. The concept about "sample" and "pool" are not well explained and seem in fact to be conceptually misleading.

Sample is a misnomer. The samples that are actually used in the data are sequence clusters. The species is clusters into groups of strains with high sequence similarity as is done in many other analyses of bacterial genomes (for example using the software POPPUNK). Each of these clusters is a putative clonal complex, i.e. a set of strains that share a recent common ancestor, relative to the rest of the sample. This is seen for example in figure 2A, where the clusters are shown in alternate colours.

Sample is a misleading name because sample refers to the process of collection of bacteria, which is not directly related to their evolutionary relationships. A sample of strains is collected over a time period or at a particular hospital or from a particular set of patients or with a particular growth media. It’s not determined post-hoc based on sequence similarity.

Pool is presented about being an idea about gene pools. But the actual gene pool for any bacteria is unknown and indeed may not have a discreate boundary. As I understand it, the entities that are in fact used as pools in this analysis are actually just designated species, which are essentially designated by a combination of traditional phenotypic criteria and (in the genomics era) sequence similarity. It is not based on any measurement of gene flow. Therefore, implying that the measurement is taken over pools is misleading.

The manuscript mentions that pools may be sets of strains isolated from particular geographic locations, but this is in fact a sample. A pool, as in a set of organisms that constitute a gene pool, is not just a collection of genomes, it is a set of organisms in nature.

[Editors’ note: further revisions were suggested prior to acceptance, as described below.]

Thank you for resubmitting your work entitled "Core genes can have higher recombination rates than accessory genes within global microbial populations" for further consideration by *eLife*. Your revised article has been evaluated by Gisela Storz (Senior Editor) and a Reviewing Editor.

The manuscript has been significantly improved with all reviewers commenting on the considerable effort that you and co-authors have made. However, there remains one issue that needs attention as outlined below in the request from Reviewer 1.

*Reviewer #1 (Recommendations for the authors):*

The correlations in figure 3 seem very strong but I am doubtful of their interpretation. It seems to me that a more direct approach than to compare different samples (which I continue to think should be called clusters) is to compare different genes. I.e. do genes with higher divergence show lower recombination? This could also help to answer the extent to which lower recombination in the accessory genome was largely a function of higher divergence levels, for example. In any case I think the method should be applied to data simulated from a recombining population with homogeneous bacterial recombination. If the method is really picking up homology dependent recombination rates, then it should give no correlation when applied to such data. Generally, simulations are very valuable for this kind of paper because they force the authors to clarify what it is they are trying to measure and what the null expectation is in the absence of any such effect.

The abstract is light on results and has too much background in it.

*Reviewer #2 (Recommendations for the authors):*

This is a greatly improved version of the paper - my main concerns regarding placing the work within a broader context of the literature have been addressed very well. My specific questions relating to the approach have bee addressed / clarified in the rebuttal and the text modified appropriately.

---

## [Author Response]

[Editors’ note: the authors resubmitted a revised version of the paper for consideration. What follows is the authors’ response to the first round of review.]

Comments to the Authors:We are sorry to say that, after consultation with the reviewers, we have decided that this work will not be considered further, at least in this version, for publication by eLife. This said, as you will see from the reviewers' comments, all recognise the importance of the subject and the need for new approaches to understand evolutionary forces shaping pan and core genomes. Your work is very much a step in the right direction. If you find it possible to deal with the requests made by the referees, we would be willing to consider a fresh submission.Specifically, on the problematic side, is seeming discord between the major claims of the paper and the data supporting these claims. This may in part stem from lack of clarity concerning methodological aspects, but as both referees point out, there are reasons for concern. And, importantly, for this work to appeal to a broad readership, it is necessary for the work to be generally accessible. Additionally, the work would benefit greatly from a more nuanced placement of the subject within context of an extensive and relevant literature. This would particularly aid motivation for the study.Reviewer #2:Homologous recombination is a key process driving the evolution of bacterial genomes, and the rate at which this occurs varies widely between different species. Whilst this variation is still not completely understood it is certainly due to a complex combination of mechanistic and ecological drivers. In this paper Steinberg and Kussel ask a slightly different question by the addressing the variation in intra-genomic rate of homologous recombination, and specifically between core and accessory genes. The use of core genes for phylogenetic and population-level inference (for example in multi-locus sequence typing) is predicated on the view that the genes encoding essential 'housekeeping' functions are under high levels of purifying selection, and are therefore assumed not to be impacted by high levels of recombination from diverged sources that might act to accelerate diversification. In contrast, accessory genes, which are more likely to encode niche-specific traits involving interactions with the external environment, may become more selectively favoured from homologous recombination leading to increased diversification. It is important to critically re-evaluate this model as evidence accumulates, both for understanding how selection shapes the genome and populations, but also for understanding the utility of different parts of the genome for phylogenetic and population genetics inference.Even before the concept of the pan and core genome was established, varying rates of recombination between different classes of genes was a subject of active research. Jain et al. (Proc Natl Acad Sci U S A 1999 Mar 30;96(7):3801-6. doi: 10.1073/pnas.96.7.3801) speculated over 20 years ago, that 'operational' (metabolic) genes experience higher rates of transfer than 'informational' genes (those involved in DNA processing) because the latter were more likely to be involved in complex protein-protein interactions (the so-called 'complexity hypothesis'). Together with the underlying rational of MLST, and the prior use of 16s rRNA as a universal marker, there remains a common wisdom that the most conserved, ubiquitous and essential genes are also those least likely to undergo recombination. For a recent discussion of the implications of the complexity hypothesis see Novick, A., Doolittle, W.F.. Biol Philos 35, 2 (2020). https://doi.org/10.1007/s10539-019-9727-6Whilst it is unfortunate that none of this broad perspective is elaborated in the paper, Steinberg and Kussel address this question on a large-scale (10 species, >100,000 genomes), and note that in many (but not all) species accessory genes tend to undergo lower rates of homologous recombination than core genes, although this trend is far from universal. To measure recombination they use an approach called mcorr which gauges recombination on the basis of the steepness of the decline in linkage between polymorphic sites as a function of genome distance. Their model allows them to distinguish between mutational and recombinational divergence, as well as estimating the percentage of each gene sequence that has been impacted by recombination. Their results are a mixed bag, with variation between species and between the different parameters within single species. However, overall the suggestion is that accessory genes may experience less homologous recombination than core genes. This trend is consistent with arguments (some of which are expressed by the authors) relating to the homogenising effect of recombination between close parental sequences, and how, in this case, recombination can reinforce purifying rather than directional selection.While the question is an important one, and the use of large genome sequences to address it is laudable, I sense that there are two many factors at play over this broad scale to draw any firm conclusions about general trends. My main concerns with the analyses are two-fold – first it is carried out on single gene alignments which severely limits the range over which recombination can be detected and quantified (this would not be so critical if all homologous recombination was by localised transformation, but transduction and conjugation also play major roles). My second concern lies in the use of a single reference genome for each species. For species with very large pan-genomes, such as *E. coli*, this effectively means that the majority of accessory genes (those that happen to be absent in the reference) will not be included. The authors do acknowledge this, and present an analysis based on a composite reference, but they reject this approach due to the poor alignments that result. This is the perennial problem with working with accessory genes – but a reference free method for detecting SNPs in accessory genes has recently been (eg Colquhoun, R.M., Hall, M.B., Lima, L. et al. Pandora: nucleotide-resolution bacterial pan-genomics with reference graphs. Genome Biol 22, 267 (2021)). There are additional potential complications relating to how the different datasets have been sampled that have not been acknowledged. In sum, the analysis should be of interest to the field as it does demonstrate the potential for addressing this fundamental question using large WGS datasets, but my sense is that the authors would have been better served to analyse one or two species in detail, taking into account all possible confounders, rather than spread themselves too thinly across such diverse datasets.

We thank the Reviewer for their comprehensive feedback on the manuscript, and we greatly appreciate their efforts in helping us improve the manuscript.

Regarding the first main concern, which is that our use of single gene alignments truncates the range over which recombination can be quantified, we have now clarified this point in the text (Lines 143-145). While our method does rely on single gene alignments, our coalescent model with recombination does not limit the size of the recombined fragment to the length of single genes. Each recombined fragment affects pairs of loci at distances *l* that are less than the fragment size, thus recombined fragments that span multiple genes contribute to the measured correlation profiles across all the affected genes. The coalescent model infers the mean fragment size by fitting the correlation profiles, and we find that mean fragment sizes may range from

~10^2^ to ~10^5^ bp. We have included some distributions of the fragment sizes predicted for the core and accessory genomes of *Streptococcus pneumoniae* to help clarify this in Figure S1.

To address the second main concern, regarding the use of a single reference genome, we have completed an analysis using both an alignment to a single reference genome and to a reference pangenome created from 81 complete genome assemblies (which encompasses the majority of the *S. pneumoniae* pangenome according to previous analysis of its pangenome). We found very similar results with both methods. In the original manuscript, we had included this comparison in Figure S2. To better facilitate this comparison, we have reformatted these data as empirical cumulative distribution functions in Figure S4. The trends for the inferred recombination parameters are very similar, which gives us confidence that the recombination rate trends for

*S. pneumoniae* hold for both alignment methods. We have re-examined the alignments, and while the alignment quality is slightly lower for pangenome alignments (as we reported originally), our results in Figure S4 show that this does not impact the results regarding recombination parameters. Furthermore, while the Reviewer’s suggestion for using a new method for reference free alignment of raw reads does indeed look promising, it also appears to be very computationally expensive; in the manuscript the Reviewer refers to, the Authors report that aligning raw reads from just 20 isolates took ~9 hours. This method would therefore be prohibitive for our analysis of ~26,000 isolates for *S. pneumoniae* alone. We revised the Results section to emphasize these points (Lines 260-275).

To focus our analysis on a detailed study of a single species, as recommended by the reviewer, we have rewritten the manuscript to highlight primarily our analysis of *Streptococcus pneumoniae*. We have removed Figure 3 from the original manuscript which contained the multi-species analysis of recombination rates and instead provide a supplemental figure (Figure S5) and a supplemental table (Table S1) briefly summarizing recombination rate trends in different species. We also moved Figure 4 (which contained another multi-species analysis) to the supplement as Figure S7. Both Figure S5 and Table S1 now simply highlight that other species either show higher or equal recombination rates in core versus accessory genes, except for *Helicobacter pylori*, for which the effect size statistic is small. We revised the text to reflect these points (Lines 276-295).

To address the concern that the “results are a mixed bag, with variation between species and between the different parameters within single species” we sought to determine whether general relationships hold across species with regard to divergence-based recombination barriers, which have been described in many bacteria. We hypothesized that due to the highly conserved *recA*-mediated homologous recombination pathway, sequence divergence should play a key role in determining recombination rates within species, and a signal of this recombination barrier should be present in the inferred recombination parameters. In the new Figure 3, we present the analysis which shows that, in all species that we analyzed, the difference in recombination coverage between accessory and core genes declines with the difference in their mutational divergences. This general relation demonstrates that, despite variation in parameters within and between species, we detect the existence of genome-wide, within-species recombination barriers in many different bacteria. We describe these results in a new section (Lines 297-327).

Lastly, to address the request that the manuscript acknowledge potential complications with how datasets were sampled, we have revised the text to explicitly discuss issues related to sampling and sample construction on Lines 172-197 and to point out how our method addresses these. We now call the reader’s attention to this discussion earlier on at Lines 126-127.

General: the paper would be vastly improved by a broader review of the literature and a more clear motivation for the study (see public response for some pointers). The introduction is frankly a bit waffly.

We thank the Reviewer for this very helpful feedback on the structure of the introduction. We have almost entirely re-written the Introduction to the paper (Lines 35-113) to better encompass literature findings and clearly motivate the study. We also appreciate the useful pointers the Reviewer gave in their public response, which helped guide this revision. We hope we have covered much of the key related literature, and if the reviewer is aware of additional papers we may have missed, we would be glad for those references as well.

Abstract:The observation that some species consist of overlapping gene pools (or even one single gene pool) to a degree simply reflects inconsistencies in our demarcation of species (for example S. flexneri is really just a clone of *E. coli*, and N. gonorrhoeae is really just a clone of N. meningitidis).

We agree with the Reviewer’s comment, and because this point was in fact tangential to our main findings, we have decided to remove the analysis of overlapping gene pool structure from the manuscript. This decision is also responsive to comments by Reviewer 3, and enables us to maintain focus throughout the text on the major results regarding the differences in recombination rates between the core and accessory genome.

Line 25 the role of population structure in genome evolution – I'm not sure about the causal direction here – it seems to make more sense the other way around to me.

We removed Line 25 as it was no longer relevant without the discussion of pool structure.

Line 29 – it has been established for a long time that core genes are under strong purifying selection. Claiming this as news somehow detracts from the aspects of the paper that are really novel.

We removed Line 29 from the Abstract as we agree that it detracts from the main points of the paper.

Line 43 – the ability to quantify rates in different parts of the genome hasn't been limited, but has been going on for a long time. For an example, see Nat Commun 2014 May 23;5:3956. doi: 10.1038/ncomms4956.

We have edited the text of the Introduction (Lines 53-57) to reference this and other work quantifying variation in recombination across different genes.

As I understand the model, it is based on the decay of linkage over distance (Figure 1C,D) but I have struggled a bit with how you move from this to the data you present in the other figures – it would be reassuring to us non-modellers if you could show some more examples of this decay to understand more clearly what the results you present are really based on.

We have added additional examples of correlation profiles in Figure 2B (line 207), and on lines 190-197 we discuss how clusters are used to measure the decaying correlation profiles, yielding the parameters distributions shown in Figure 2C-E.

Are recombination events, like synonymous SNPs, also assumed to be neutral?

There is no implicit assumption of neutrality for recombination events in the model. We have added text to clarify this (Lines 134-135), and thank the Reviewer for this question as we now realize this key point may not have been clear.

Line 120 – explain w/n.

As we have removed the analysis of pool structure from the manuscript, we no longer use the terms ‘w/n’ and ‘btwn’ in the manuscript.

Line 136 – meaning 'no two sequences within a cluster'?

That is correct, and we have amended this line in the text correspondingly (presently on Lines 192-194).

Figure 2B – why not separate points for within and between clusters?Figure S2 – why exclude the marginal plots?

We have changed Figure 2 to just show empirical cumulative distribution functions (*ECDF*s) of the recombinational and mutational divergence (Figure 2C-D in the revised manuscript). We chose to not separate the points for within and between clusters in the revised manuscript because we have removed the analysis of pool structure, for which the distinction of within and between clusters was important. Figure S2 has been replaced with Figure S4, which now just shows *ECDF*s of recombination parameters for the different alignment methods side-by-side, which we hope allows for a clearer comparison between the results using an alignment to a single reference genome versus a pangenome reference genome.

Line 194 – There is certainly an effect of codon bias on substitution rate – the fact the experimental evolution data hasn't picked this up is more a comment on these experiments.

We agree with the Reviewer’s comment, and have revised the text to estimate the magnitude of the effect in *S. pneumoniae* and relate it to our analysis and to the prior results in experimental evolution. We moved this section to the Discussion (Lines 403-411).

Line 264 – the P values presented in S table 1 are two-signed – which ones indicate higher rates of recombination in the core genes?

We have amended Table S1 so that it also includes the median recombination rates and effect size statistics (Cohen’s *d*) quantifying the difference in rates between the core/accessory genes. Values of *d* < 0 indicate higher rates of recombination in core genes.

Figure 4 – 60% of the core gene sequences in S. enterica have experienced recombination? This seems a lot -how does this figure (and other species specific data) compare with what is in the literature? Could the authors comment on the fact that in *E. coli* (and S. flexneri) the strict core genes show less recombination than the cloud core.

We checked the literature with regard to *S. enterica*, and note that another recent study using a smaller dataset shows comparable levels of recombination in the core genes (see Park and Andam, *mSystems* 5:e00515-19, 2020; doi:10.1128/mSystems.00515-19). While other studies have examined recombination across housekeeping and core genes, there is scant existing data for accessory genes. In particular, no studies to our knowledge have presented direct comparisons between core and accessory genes. We have revised the text to address these points (Lines 385-388). We have also added discussion commenting on potential reasons why the cloud genes for *E. coli* and *S. flexneri* show higher recombination rates than core genes (Lines 358361). We would also like to note that Figure 4 has been moved to the supplement as Figure S7.

Line 322 'complicates' not 'complexifies'.

Line 322 has been removed as it refers to the analysis of pool structure, which is not included in the revised manuscript.

Line 332 – 340 this model isn't really novel.

We have amended the text preceding this statement to reference research which has shown that genes with lower sequence divergence have a reduced barrier to homologous recombination and modelling results which have suggested that core genes experience more homologous recombination leading to homogenization (Lines 420-429).

Reviewer #3:This paper presents interesting results comparing core and accessory genome evolution.The headline result, if correct, is both important and reasonably easy to understand; namely that in Pneumococci and other bacterial species, on average homologous recombination rates (or more precisely the effect of recombination in reassorting sequence diversity) are higher in core genome elements, while diversity is higher in accessary genome elements.This is an interesting result and is buttressed by showing some dependence on gene frequencies. It is not entirely surprising, since core genes have more opportunities for recombination due to being present in every strain, while accessory genome elements have more opportunity for being imported from other species. Nevertheless, a quantitative result is important enough to be newsworthy, even though it does not do an enormous amount to elucidate the mechanisms responsible for the difference. This would require more explicitly phylogenetic methods or a more in-depth analysis of the difference between species, which is currently superficial.

We thank the Reviewer for their comprehensive feedback on the manuscript, and for highlighting the broad interest of this work to the community. We greatly appreciate their efforts in helping us improve the manuscript.

To address the Reviewer’s comment regarding connecting the quantitative result on the difference in recombination rates between core and accessory genes to underlying mechanisms, we have performed additional analysis which is presented in a new Figure 3 and in Results (Lines 298-328). Specifically, we tested whether divergence-based recombination barriers are related to differences between core and accessory genes’ rates of recombination, across clusters within each species. We find pronounced negative correlations in the paired differences of sequence divergence and recombination coverage for accessory and core genes in *S. pneumoniae* (Figure 3A), and in the other bacteria that we have analyzed (Figure 3B,C). We include this general finding as a first major step toward elucidating the mechanisms underlying the quantitative differences we have presented.

However, I am not sure I am persuaded by the second part of the result, since looking at supplementary figure 4, it is not obvious to me that there is a clear trend for recombination parameters, core and accessory genomes seem to give very inconsistent results in different species, for which no real explanations are given.

To address the Reviewer’s concern regarding the generalizability of our results to multiple species, we have performed two statistical analyses on the relative recombination rates, shown in Figure S4 and Table S1.

First, we computed Cohen’s *d* statistic showing the effect size for the difference in relative recombination rates (Figure S4, Table S1). This shows a non-zero effect size in 9 out of 12 species, and in all but one of these (*H. pylori*), we find that core genes have higher recombination rates than accessory genes (*d* < 0). Second, we list the median relative recombination rates for core and accessory genes in each species (Table S1). We find that in 10 out of 12 species the medians are significantly different (within 95% confidence intervals), and among these in all but *H. pylori* and *N. gonorrhoeae* the core genes have higher recombination rates than accessory genes. Thus, the general trend according to these two independent metrics is quite consistent. Additionally, in the exceptional cases of *H. pylori* and *N. gonorrhoeae*, the *d* statistic is small in magnitude, indicating the overall effect size is small.

In response to both Reviewers’ feedback we have decided to focus most of the results on the analysis of *Streptococcus pneumoniae*, which serves as the main case study of the paper. We then present a very focused set of results on the additional bacterial species in order to test the generality of the results that we found in *S. pneumoniae*. In addition to Figure S4 and Table S1, the new Figure 3 (discussed in the previous comment) addresses the reviewer’s concern regarding explanation of the observed trend in recombination parameters between core and accessory genes.

There is one important technical check that should be carried out, which is that core genes should be artificially thinned to match the frequency of accessory gene elements and then the analysis repeated, to see whether this has any effect on in the inference. Indeed, because the alignment method is shown to have a considerable effect on the inference, especially of recombination parameters (supplementary figure 2) this thinning would be ideally done before alignment, although I recognize that this would be time-consuming to implement.

We have carried out the technical check suggested by the Reviewer – artificially thinning the core genome – and found similar results for the “thinned” core genome compared to the actual core genome, suggesting that the difference in sequence pairs between core and accessory genome does not affect parameter inference enough to change our interpretations. We provide these data as Figure S3, and have amended the text (Lines 253-259). We note that we performed this control post-alignment because, as we have discussed in our response to Reviewer #2, the distributions of recombination parameters inferred from the different alignment methods are similar, and we have provided a new supplemental figure (Figure S4, described in the response to Reviewer #2’s question) which we hope better highlights the similarity in parameter inference from different alignment methods.

Figure 3 is presented as the results from 12 species but it’s really hard to judge what this figure actually tells us. Agglomorating different species into single graphs seems meaningless and makes it impossible to see what is actually going on with individual species.

We agree with the Reviewer’s comment on the difficulty of comparing and assessing multiple species on a single graph. In response to this comment, as well as the comments of Reviewer #2, we focus most of our analysis throughout the main text on a detailed study of *Streptococcus pneumoniae*. We have removed Figure 3 from the original manuscript which contained the multi-species analysis of recombination rates and instead provide a supplemental figure (Figure S5) and a supplemental table (Table S1) briefly summarizing recombination rate trends in different species. We also moved Figure 4 (which contained another multi-species analysis) to the supplement as Figure S7. Both Figure S5 and Table S1 now simply highlight that other species either show higher or equal recombination rates in core versus accessory genes, except for *Helicobacter pylori*, for which the effect size statistic is small. We revised the text to reflect these points (Lines 276-295).

We hope that with these changes the interpretation of the main results is much clearer.

Aside from questions about to what extent the data really supports the headline result, my main problem with the paper is that the analysis underlying the result and in particular the meaning of the figures is difficult to understand. As it stands, I do not think it is suitable for a broad readership. The concept about "sample" and "pool" are not well explained and seem in fact to be conceptually misleading.

We thank the Reviewer for bringing these points regarding clarity and suitability for a broad readership to our attention. To address these, we have simplified Figure 1 so that it no longer includes the analysis of pool structure, which was tangential to our central results. We now focus the figure entirely on inferring recombination parameters from correlation profiles and explaining the model. We have added additional example correlation profiles of synonymous substitutions into Figure 2 to better illustrate how we go from sequencing data to recombination parameters. We also removed the analysis of pool structure from the text of the manuscript and from Figures 1-2 (as well as Figures S1 and S3).

We have re-written the Introduction almost entirely in order to better place the paper into context for a broad readership, and to introduce the major concepts and methods more clearly for this audience. We have also revised the Results section to clearly explain the issues involved in the concept of “sample” and “pool”, (Lines 122-127) and particularly in terms of sample construction (Lines 172-197), discussed further in the next comment.

We hope that these changes have substantially improved the readability and suitability of the manuscript for a broad readership.

Sample is a misnomer. The samples that are actually used in the data are sequence clusters. The species is clusters into groups of strains with high sequence similarity as is done in many other analyses of bacterial genomes (for example using the software POPPUNK). Each of these clusters is a putative clonal complex, i.e. a set of strains that share a recent common ancestor, relative to the rest of the sample. This is seen for example in figure 2A, where the clusters are shown in alternate colours.Sample is a misleading name because sample refers to the process of collection of bacteria, which is not directly related to their evolutionary relationships. A sample of strains is collected over a time period or at a particular hospital or from a particular set of patients or with a particular growth media. It’s not determined post-hoc based on sequence similarity.

We thank the Reviewer for bringing this point to our attention, which we hope we have now been able to fully clarify. We understand that the term “sample” as it was previously described in the manuscript may have been misleading. By “sample” we do not mean to refer to the process of collection itself, but instead we are using the term in its statistical sense, referring to the process of statistically sampling from a larger distribution; in this case, sampling from the bulk gene pool. In our analysis, the sequence clusters we use “sample” from these bulk gene pools. We have made this point explicit in the text (Lines 173-175), and we have added additional text about the role of clustering in our sample construction (Lines 175-197).

Pool is presented about being an idea about gene pools. But the actual gene pool for any bacteria is unknown and indeed may not have a discreate boundary. As I understand it, the entities that are in fact used as pools in this analysis are actually just designated species, which are essentially designated by a combination of traditional phenotypic criteria and (in the genomics era) sequence similarity. It is not based on any measurement of gene flow. Therefore, implying that the measurement is taken over pools is misleading.The manuscript mentions that pools may be sets of strains isolated from particular geographic locations, but this is in fact a sample. A pool, as in a set of organisms that constitute a gene pool, is not just a collection of genomes, it is a set of organisms in nature.

The Reviewer makes an excellent point, and we have sought to clarify this important aspect of the paper. We agree with the Reviewer that the gene pool is unknown and may not have discrete bounds. We recognize that the original manuscript may not have adequately described what our method assigns to be the pool. In fact, our method does not use or designate any entity as the pool, but instead we use the analyzed sequences (i.e., the “sample”) to infer recombination parameters of the greater, unobserved gene pool with which the sample recombines. The pool parameters are distinct from those of the sample (the sample being the set of sequenced genomes used to compute each correlation profile, e.g., a single sequence cluster). We revised the text to clarify these points (Lines 122-127; 172-177).

We also noticed that Line 108 in the original manuscript may have been confusing, as it read: “Samples may have access to different sequence pools, e.g., due to geographic constraints or differences in host niches.” We recognize that the way this was written may have been unintentionally misleading; we meant that samples could be constrained biogeographically, which may lead to access to different gene pools, not that the gene pools are constrained. We have removed this line from the manuscript, and amended the text to further clarify this point (Lines 177-197).

[Editors’ note: what follows is the authors’ response to the second round of review.]

The manuscript has been significantly improved with all reviewers commenting on the considerable effort that you and co-authors have made. However, there remains one issue that needs attention as outlined below in the request from Reviewer 1.Reviewer #1 (Recommendations for the authors):The correlations in figure 3 seem very strong but I am doubtful of their interpretation. It seems to me that a more direct approach than to compare different samples (which I continue to think should be called clusters) is to compare different genes. I.e. do genes with higher divergence show lower recombination? This could also help to answer the extent to which lower recombination in the accessory genome was largely a function of higher divergence levels, for example. In any case I think the method should be applied to data simulated from a recombining population with homogeneous bacterial recombination. If the method is really picking up homology dependent recombination rates, then it should give no correlation when applied to such data. Generally, simulations are very valuable for this kind of paper because they force the authors to clarify what it is they are trying to measure and what the null expectation is in the absence of any such effect.

We agree with the Reviewer that multiple effects could determine the homology-based recombination barriers examined in Figure 3 of the previous version of the manuscript, and that simulations may play an important role in revealing these dependencies.

As the modeling work needed to fully address this point is extensive, given the complex population structure underlying these datasets, we have removed the original Figure 3 from the manuscript and corresponding text (Lines 27-32, 90-101, 108-110, 299-327, and 423-426 in the previous version).

The revised manuscript is thus focused on (i) the major result showing that core genes can recombine more frequently than accessory genes, and the related findings (ii) that mutational divergence is lower in core genes, and (iii) that gene frequency correlates with recombination rate.

We moved the previous Figure S5 to be the new Figure 3, showing the key statistical analysis supporting the differences in the recombination rates of core and accessory genes. We have also added a part B to the new Figure 3, which displays effect sizes for the pool’s mutational divergence, and we include additional statistics to support these results in Supplementary File 2. In the revised manuscript, we have added a description of the pool’s mutational divergence results on lines 242-247 and include a brief discussion of their potential relationship to the observed recombination rate differences on lines 328-337 of the revised manuscript.

The results presented in the new Figure 3 are consistent with the Reviewer’s comment above regarding divergence levels and recombination rates. To confirm this, we conducted the additional control that the reviewer suggested, i.e., to test if genes with higher divergence show lower recombination. Using the *S. pneumoniae* dataset shown in Figure 2, we binned genes by their synonymous diversity into low or high diversity bins (i.e., lower 50%, upper 50%), and inferred recombination rates separately in each bin for all clusters and cluster pairs. The results are shown in the bar plot in Author response image 1: genes with higher diversity (and, correspondingly, divergence) show lower recombination rates (the y-axis shows the median pool recombination rates, and the error bars are 95% bootstrap confidence intervals).

**Author response image 1. sa2fig1:** 

We leave further detailed analysis on the role of homology barriers to future work.

The abstract is light on results and has too much background in it.

We agree with the Reviewer’s comments, and have added additional text to the Abstract describing key results on Lines 24-31 of the revised manuscript. To remove some of the background, we took out Lines 17-19 of the original Abstract.

We would like to thank the Reviewer for taking the time to carefully read the manuscript, and for their insightful feedback which has further improved the manuscript.